# Impact of SARS-CoV-2-Related Hygiene Measures on Community-Acquired Respiratory Virus Infections in Lung Transplant Recipients in Switzerland

**DOI:** 10.3390/medicina59081473

**Published:** 2023-08-16

**Authors:** Isabelle Baumann, René Hage, Paola Gasche-Soccal, John-David Aubert, Macé M. Schuurmans

**Affiliations:** 1Faculty of Medicine, University of Zurich, 8032 Zurich, Switzerland; isabellejessica.baumann@uzh.ch (I.B.);; 2Division of Pulmonology, University Hospital Zurich, 8091 Zurich, Switzerland; 3Division of Pulmonology, University Hospitals Geneva, 1205 Geneva, Switzerland; 4Division of Pulmonology, University Hospital Lausanne, 1011 Lausanne, Switzerland

**Keywords:** lung transplant recipients, COVID-19, hygiene measures, CARV, virus infections

## Abstract

*Background and Objectives*: Community-acquired respiratory virus (CARV) infections pose a serious risk for lung transplant recipients (LTR) as they are prone to severe complications. When the COVID-19 pandemic hit Switzerland in 2020, the government implemented hygiene measures for the general population. We investigated the impact of these measures on the transmission of CARV in lung transplant recipients in Switzerland. *Materials and Methods*: In this multicenter, retrospective study of lung transplant recipients, we investigated two time periods: the year before the COVID-19 pandemic (1 March 2019–29 February 2020) and the first year of the pandemic (1 March 2020–28 February 2021). Data were mainly collected from the Swiss Transplant Cohort Study (STCS) database. Descriptive statistics were used to analyze the results. *Results*: Data from 221 Swiss lung transplant cohort patients were evaluated. In the year before the COVID-19 pandemic, 157 infections were diagnosed compared to 71 infections in the first year of the pandemic (decline of 54%, *p* < 0.001). Influenza virus infections alone showed a remarkable decrease from 17 infections before COVID-19 to 2 infections after the beginning of the pandemic. No significant difference was found in testing behavior; 803 vs. 925 tests were obtained by two of the three centers during the respective periods. *Conclusions*: We observed a significant decline in CARV infections in the Swiss lung transplant cohort during the first year of the COVID-19 pandemic. These results suggest a relevant impact of hygiene measures when implemented in the population due to the COVID-19 pandemic on the incidence of CARV infections.

## 1. Introduction

Community-acquired respiratory virus (CARV) infections are frequent in the general population and lung transplant recipients (LTR) [1,2,3]. Often leading to self-limiting upper respiratory tract infections, these viruses may cause serious complications and can harm the long-term outcome of LTRs. CARV infections frequently lead to an acute lung allograft dysfunction (ALAD), which is characterized by lung function decline, which, in some cases, may never recover [4]. Although many studies are inconclusive on the potential triggers of acute rejection episode, there is increasing evidence that such CARV infections are strongly associated with the development of chronic lung allograft dysfunction (CLAD) [1,3,4,5,6,7,8,9,10,11,12,13,14]. LTRs are also considered to be more vulnerable than other solid organ transplant recipients (SOT), most likely because the transplanted organ is in direct contact with the environment and is directly affected by these infections [15]. Additionally, the level of immunosuppression required to reduce rejection rates is more pronounced than in most other solid organ transplant recipients. Therefore, preventing CARV infections in LTRs is paramount to avoid acute and chronic allograft dysfunction. Indeed LTRs have regularly been instructed by their physicians about hygiene and ways to lower the risk of contracting infections [16]. Furthermore, transplant physicians have a low threshold for performing PCR-based diagnostic tests, which allows for the early modification of immunosuppressive treatment in an LTR upon diagnosing a CARV infection [17].

When the COVID-19 pandemic hit Switzerland in 2020, the Swiss public implemented government-imposed hygiene measures, such as hand hygiene, social distancing, and face masks [18]. We hypothesized that this raised awareness and implementing general hygiene measures would positively affect the occurrence of CARV infections within the Swiss lung transplant cohort. Thus, we postulated that there would be a higher incidence of CARV infections in the year before the COVID-19 pandemic (first period assessed) compared to the first year of the COVID-19 pandemic (second period assessed). 

## 2. Materials and Methods

### 2.1. Study Design, Inclusion, and Exclusion Criteria

We conducted a retrospective, multicenter study on lung transplant recipients from three lung transplant units in Switzerland: the University Hospital of Zurich (USZ), the University Hospital of Lausanne (CHUV), and the University Hospitals of Geneva (HUG). 

The periods of interest were the year before the COVID-19 pandemic, defined as 1 March 2019 to 29 February 2020, and the first year of the COVID-19 pandemic, defined as 1 March 2020 to 28 February 2021.

We included all LTRs that were alive on 28 February 2021 (study ending) and transplanted between 2008 and 28 February 2019. Patients who received combined organ transplantation (for example, heart and lung) and pediatric patients (<18 years old on 28 February 2021) were excluded. In addition, all patients who received a lung transplant after the study began on 28 February 2019 were excluded. 

The data collection occurred with the help of the Swiss Transplant Cohort Study (STCS) database: a prospective observational cohort that collects data from Swiss solid organ transplant recipients. In addition, other necessary data were retrieved from local transplant centers’ databases.

All patients provided written informed consent, and the local Ethics Committees of each center approved data collection. The Ethics Committee of the University Hospital of Zurich approved the protocol of the current study with the protocol number 2021-00665. 

### 2.2. Data Collection

The STCS database provided encoded data on patients meeting the inclusion criteria for the two time periods. The variables included age and sex, underlying disease leading to transplantation, all viral infections detected within the first three years after transplantation and viral infections leading to hospitalization after the first three years.

To complete the data, one investigator (IB) reviewed medical records at individual sites (USZ, CHUV, HUG) and collected data on flu vaccination status, number of nasopharyngeal swabs, and number of bronchoalveolar lavage (BAL) procedures. Additionally, data on the viral infections of patients who had received their lung allograft more than three years ago were compiled since the STCS database only collected detailed data for the first three years post-transplant and for all hospitalizations. 

Patients followed up in other Swiss University hospitals (Basel and Berne) were excluded from the analysis due to incomplete data unless their data were available in one of the three participating centers (local patient records). 

### 2.3. Definitions

CARV infection was defined as the detection by the multiplex PCR of Adenovirus, human (h) Bocavirus, Coronavirus (229E, HKU1, NL-63, OC43), Metapneumovirus, Influenza virus A + B, Parainfluenza virus (1–4), Rhino- and Enterovirus, Respiratory Syncytial Virus (RSV) A + B, and SARS-CoV-2 (only in period 2) in a naso-, pharyngeal swab, or bronchoalveolar lavage [19].

The concurrent detection of more than one virus was counted as one viral infection episode, but when looking at the viral infections per type of virus, each virus was counted separately. 

We defined prolonged viral shedding as the duration of detectable viral material of the same virus in consecutive PCR tests measured from the first to the last detection (at least 14 days apart) unless separated by an asymptomatic phase of at least 4 weeks. If this criterium of an asymptomatic phase of 4 weeks was not fulfilled, the infection was considered the same viral infection. 

### 2.4. Outcomes

The primary outcome of our study was the annual incidence of CARV infections. The incidence of respiratory infections was calculated for both periods. The secondary outcome was the type of CARV infection, with a special focus on the incidence of Influenza virus infections, considering the Influenza vaccination status as a possible confounder. 

### 2.5. Statistical Analysis

We used descriptive statistics, including binomial testing, to illustrate the characteristics of the patients included in the study and our general results. Data analyses were performed using EXCEL (Microsoft) and IBM SPSS Statistics, version 27. Differences were considered statistically significant at *p* < 0.05. Two-sided significance tests were performed. 

## 3. Results

### 3.1. Patient Characteristics

A total of 221 lung transplant recipients from three Swiss transplant centers were included in our study. The mean age at the beginning of the study was 50.5 years, and 49% of the patients were female. Patients underwent lung transplantations with a mean of 3.9 years before study entry. The most common disease leading to lung transplantation was cystic fibrosis (33.5%), followed by chronic obstructive pulmonary disease (29%). The clinical and epidemiological data of all included patients are presented in Table 1. 

### 3.2. Test Behavior: Before and during the COVID-19 Pandemic

The number of samples per period is only known from two testing centers, USZ and HUG, Table 2. Testing in all centers is based on clinical suspicion, not routine sampling.

Combining these two periods, 1728 nasopharyngeal, pharyngeal, nasal swabs and 68 bronchoalveolar lavages were performed on 150 transplant recipients in the two centers.

Because SARS-CoV-2 as a viral pathogen was not yet known in the first period, there was no COVID-only testing. Therefore, to make a fair comparison, we first excluded COVID-only swabs from the second period, leaving us with similar numbers of tests carried out in the two periods: 803 and 779, respectively. This observation could also be confirmed by the number of tests undertaken per patient; on average, a patient was tested 5.7 times in the first period, while in the second period, it was 5.3 times. When including COVID-only swabs, 925 tests were carried out in the second period, resulting in an average of 6.3 tests per patient.

COVID-only swabs performed in the second period accounted for 15.8% (146 of 925). 

### 3.3. Epidemiology of CARV Infections

Overall, 228 CARV infections (including SARS-CoV-2) were diagnosed during these two periods, as summarized in Table 3.

A total of 157 CARV infections were described among 105 (47.5%) of 221 patients followed up in the first period, indicating that some patients had more than one infection. Additionally, 9 of 157 infections were coinfections with one or more virus species found in the PCR testing. In the second period, 71 infections, including COVID-19 (*n* = 19), were diagnosed in 58 (26.2%) patients. No coinfections were diagnosed during this period. In the first period, prolonged viral shedding was observed in 30 (19.1%) of the 157 infections, compared to 13 (18.3%) of the 71 infections in the second period. We observed a significant reduction of 54% (*p* < 0.001) in CARV infections from the first to the second period. 

Rhino- and Enteroviruses were the most frequently encountered viruses making up more than half of all the viruses found in both periods, with a total of 53.0% vs. 60.6%, respectively. The most common reason for hospitalization was a Parainfluenza virus infection since three hospitalized patients had coinfection with Parainfluenza virus or the infection alone. No hospitalizations related to respiratory viral infections were reported during the second period.

### 3.4. Influenza Virus

Overall, 19 Influenza virus infections were diagnosed in both periods, with a ratio of Influenza A vs. Influenza B of 17 vs. 2. Considering Influenza virus infections per period, a pronounced decline in the second period was observed. At the same time, vaccination rates were comparable in both periods (71.5% vs. 65.2%), as shown in Table 4. 

### 3.5. Excluded Patients

In total, 34 patients were excluded from the analysis due to incomplete data (i.e., follow-up in other centers, relocation, etc.).

During the study period, 35 patients died and were excluded from our analysis. In total, 4 out of the 35 patients died of infectious disease, but none died of a disease associated with a COVID-19 infection. 

Additionally, 67 patients were transplanted during our study period and were excluded due to the short follow-up duration, as shown in Table 5.

## 4. Discussion

This retrospective study beginning one year before and including the first year of the COVID-19 pandemic, suggests that the implemented hygiene measures led to a statistically significant decrease in CARV infections, particularly Influenza virus infections, in lung transplant recipients. The most frequently encountered viruses in both periods were the Rhino-/Enteroviruses. Furthermore, testing behavior was only minimally influenced by the COVID-19 pandemic, making our cohort ideal for investigating the impact of hygiene measures on the incidence of CARV infections. Finally, the CARV infections in our cohort were mostly mild, with no deaths reported and only a few hospitalizations necessary.

As hypothesized at the beginning of the study, we found a statistically significant reduction in CARV infections from the first to the second period. Although COVID-19 infections only occurred in the second period, the reduction in CARV infections, although smaller, was still statistically significant. This observation is likely explained by the implementation of public hygiene measures in the population by the Swiss Government in the second period. On the 16 March 2020, the Swiss Government ordered a lockdown that was upheld until the 27 April. Subsequently, some measures were partly downscaled gradually until June 2020. During the whole year of 2020, measures such as wearing a mask on public transport and inside public buildings, restrictions on the number of people for private and public events, the shutdown of dance clubs, and restricted opening hours for restaurants and bars were implemented on a large scale [18]. Additionally, throughout the pandemic, instructions on hand hygiene were spread, and recommendations to work from home and social distancing were promoted, leading to a broad awareness of hygiene measures in the Swiss public. 

In this study, we did not assess the impact of each specific hygiene measure on the spread of CARV. However, we assumed that the lockdown, later mask use, and social distancing must have had the greatest impact, this also being the main difference to the pre-pandemic hygiene instructions already known to LTRs.

To date, one of the biggest analyses on the effectiveness of physical interventions was the systematic review by Jefferson et al., which, in contrast to our study, concluded that there was no sound evidence that the use of face masks or implemented hand hygiene could interrupt or reduce the spread of respiratory viruses [20]. It is important to point out that all the studies included in that review were conducted in non-epidemic periods and, therefore, might not represent our research context. Especially since the pandemic spread of SARS-CoV-2 and its impact on healthcare systems, as well as the measurements taken by governments, which are unique in their magnitude, these far outweigh the effect of the measures taken for the H1N1 pandemic one decade earlier. 

In the context of the COVID-19 pandemic, many other studies have investigated the effect of hygiene measures on CARV infections [21,22,23,24,25,26,27,28,29,30]. Only three further investigations have analyzed the impact of COVID-19-related measures on lung transplant recipients’ infection rates [4,31,32]. We think lung transplant recipients are an ideal cohort to investigate when considering the effects of government-imposed hygiene measures on CARV infections. First, CARV infections pose a big threat to lung transplant recipients, with possible lung function decline in the case of ALAD and a strong association with the development of CLAD [1,4,7,12,14]. Thus, preventing such infections is paramount for these patients. This is mirrored by the fact that lung transplant recipients were instructed by their physicians long before the COVID-19 pandemic on the importance of hygiene measures, especially ways to lower the risk of CARV infection. Such instructions mainly contained information on the importance of hand hygiene and the encouragement to avoid crowds, and if avoidance thereof was impossible, the recommendation to wear masks. Thus, in the wake of the pandemic, the public drastically changed their behavior and newly implemented similar and even broader hygiene measures than had been previously recommended to LTRs.

Second, the testing behavior of LTR was already well and uniformly established at the beginning of the pandemic: In all three transplant centers in Switzerland, LTRs were instructed to immediately call their transplant center when experiencing symptoms and/or measuring a decrease in home spirometry values (a loss of lung function) [17]. This often led to an emergency visit where a nasopharyngeal swab was taken, or further investigations, such as a bronchoscopy, were initiated in unclear situations. A multiplex PCR test was used to identify CARV infections. This procedure aligned with general recommendations and was valid for many years before the COVID-19 outbreak [33,34,35]. The results of our investigation confirm this, as we found that testing behavior was similar in both periods and was hardly impacted by the COVID-19 pandemic: the year before the pandemic, an average of 5.7 tests per patient were being performed, while in the first year of the COVID-19 pandemic, it was 5.3, respectively, 6.3 tests per patient, whereby the second number also considers the COVID-only swabs. 

Other studies investigating the impact of non-pharmaceutical measures on CARV infections support our findings and show that implementing hand hygiene, the use of face masks, and social distancing were associated with a decrease in CARV infections in their cohorts [21,22,23,24,25,26,27,28,29,30]. Most of these studies investigated the change in the number of emergency visits in the general population, which they concluded on the number of infections and, thus, the impact on CARV infections. It is known that emergency visits have declined drastically due to restrictions such as lockdowns and the general recommendation only to seek medical care for more severe emergencies. Moreover, some patients avoided contact with healthcare institutions for fear of contracting SARS-CoV-2: an additional factor that reduced healthcare usage. Therefore, several CARV infections in these studies might have been missing, leading to a bias. By having similar testing behaviors in both periods due to the high adherence of LTR, as elucidated above, we could exclude the possibility that different testing behaviors caused a decrease in infections. With similar testing and consistent behavior regarding hygiene measures by our cohort in both periods, the main difference between the two periods remains the government-imposed COVID-19-related measures. Therefore, this could be interpreted as the main driving force behind the decline in CARV infections. This is supported by three other investigations on LTR cohorts during the COVID-19 pandemic with similar conclusions [4,31,32].

Regarding infection epidemiology, Rhino-/Enteroviruses were encountered most frequently throughout the two years. This was expected when comparing the results with those of other studies that examined the epidemiology of CARV [6,10,36,37]. The fact that Rhino-/Enteroviruses were also the most common viruses during the COVID-19 pandemic was an unexpected finding. Other studies have found that Rhinoviruses and Adenoviruses were the least affected by the COVID-19 pandemic [26,38]. Whether this means that hygiene measures do not impact the transmission of Rhino- and Enteroviruses needs to be evaluated in further studies. 

We did not investigate the seasonality of the viruses. Nevertheless, we know that seasonality is a possible confounder when examining the number of respiratory infections. Therefore, we chose to investigate two full years to ensure that every season was represented in both periods, thus accounting for possible seasonal variations as confounders. 

Since we found a clear decline in the transmission of Adenovirus, Coronaviruses, Metapneumovirus, Parainfluenza Virus, and RSV in the second period, our study suggests that these viruses are susceptible to implemented hygiene measures.

Of note, the Influenza virus seemed especially impacted by the COVID-19 pandemic hygiene measures: in the year before the pandemic, we diagnosed 17 infections with the Influenza virus, compared to only two Influenza virus infections during the pandemic. This coincides with the general decline in Influenza virus infections in the Swiss population, as published by the Swiss Government [39]. The impact of the Influenza virus vaccination seems negligible, as the vaccination rates were similar in the two periods. 

Our findings align with several other investigations showing a significant decrease in Influenza virus cases worldwide [22,38,40,41,42,43,44,45,46,47,48,49,50,51]. These reports have similar conclusions that the hygiene measures implemented by governments appear to be the main reason for the pronounced decline in Influenza virus transmission. This observed reduction in infections is in line with preventive measures, which have been investigated in the past to stop the spread of the Influenza virus; these hygiene measures are almost identical to the hygiene measures now implemented by governments worldwide [52,53]. Nevertheless, the extent to which the Influenza virus was suppressed during the first year of the COVID-19 pandemic remains surprising. 

Several investigators have studied the impact of hygiene measures on respiratory diseases in children, including conditions other than CARV infections. They all found a decline in hospitalization and infection rates directly related to hygiene measures [54,55,56,57].

Despite being beyond the scope of our study, it is interesting to note that a decline in bacterial infections was also observed during the COVID-19 pandemic [24,41,58].

We have no information on the symptoms or severity of the diagnosed infections. However, we do know that in the first year, seven hospitalizations occurred because of CARV infections. In the second year, none of the patients in our cohort had to be hospitalized for CARV infections, not even patients with SARS-CoV-2. Remember that we excluded all patients who underwent transplantation within our study period and possibly missed some recently transplanted patients who had to be hospitalized. In addition, we are aware of several hospitalizations due to SARS-CoV-2 that occurred after our study period, as we need to remember that the pandemic continued. When looking at the patients who died during the study period and were, therefore, excluded, we know that only 4 of these 35 patients died because of an infection.

Even though prolonged viral shedding (PVS) is known to be a common problem in immunocompromised patients, such as LTR, only a few studies have investigated this. Accordingly, there is no uniform definition for prolonged viral shedding in recipients of solid organ transplants (or hematopoietic stem cell transplants) [5,59,60]. This is still true, even though more research has been conducted on PVS during the COVID-19 pandemic with variable definitions and results [61,62,63,64]. We defined prolonged viral shedding as the duration of detectable viral material of the same virus in consecutive PCR tests when measured from the first to the last detection (at least 14 days apart) unless separated by an asymptomatic phase of at least 4 weeks. If this criterium of 4 weeks for the asymptomatic phase was not fulfilled, the infection was considered the same viral infection. 

A limitation of our study is that information on the criteria by which PCR testing was initiated, for example, exact details on how many patients showed symptoms when tested, how many showed a lung function decline, etc., were missing. However, we know that the three centers used testing protocols that aligned with the American Society of Transplantation Infectious Diseases Community of Practice Guidelines and that they saw transplant patients at regular intervals, and in symptomatic patients, multiplex PCR testing was generally performed. Additionally, we know that the testing practice was not limited to the COVID-19 pandemic but had been established before. 

Another limitation was the occurrence of COVID-only tests during the second period. In some cases, COVID-only testing was performed with a multiplex PCR that did not yet include SARS-CoV-2; in other cases, it was decided only to take a COVID test. As we do not have any information on the decision process behind the tests taken, we can only assume that in the cases where only COVID tests were performed, patients were tested because of their contact with a COVID-positive person. Therefore, there was no need to test for other viruses. We might have included a possible bias by including patients in the study that were transplanted until the day immediately before our study’s initiation. As in the first year after transplantation, the risk of contracting an infection is known to be higher, possibly leading to a higher number of infections in the first period of our study. Thirty-two patients underwent transplantation in the year before our study’s initiation.

A further limitation is that 34 patients, after being followed up at University Hospital Bern and Basel, were not included due to incomplete data at the three sites investigated. 

One of the main strengths of our study was the size and characteristics of our LTR cohort from three university hospitals in Switzerland. Our cohort is representative of lung transplant recipients in many parts of the world, as key characteristics such as age, diagnosis leading to lung transplantation, and time since lung transplantation were comparable to those of lung transplant cohorts in other countries. Additionally, as elucidated above, testing behavior was not impacted by the COVID pandemic and its regulations. This excluded a possible bias and supported our assumption that the decline in CARV infections in the second year was due to widely implemented hygiene measures. This makes our cohort ideal for investigating the COVID-19-related hygiene measures’ impact on the incidence of CARV infections. 

Finally, a multiplex PCR test was performed in most of our cases; therefore, it is unlikely that a significant number of viral infections were missed in our cohort. 

## 5. Conclusions

Our study observed a significant reduction in CARV infections during the first year of the COVID-19 pandemic in the Swiss lung transplant cohort. This suggests that hygiene measures broadly implemented during the COVID-19 pandemic significantly impacted the incidence of community-acquired respiratory viral infections in lung transplant recipients.

## Figures and Tables

**Table 1 medicina-59-01473-t001:** Patient characteristics *n* = 221.

Female/Male, *n* (%)	109/112 (49/51)
Age at beginning of study (years), median (range)	55 (18–72)
Diagnosis leading to lung transplantation, *n* (%)	
Cystic fibrosis	74 (33.5)
Chronic obstructive pulmonary disease	64 (29)
Idiopathic pulmonary fibrosis	21 (9.5)
Interstitial Lung Disease	16 (7.2)
Pulmonary arterial hypertension	9 (4.1)
Others (i.e., AAT, LAM, BCT)	37 (16.7)
Time since lung transplantation, *n* (%)	
<1 year	32 (14.5)
1–3 years	48 (21.7)
3–5 years	50 (22.6)
5–10 years	84 (38.0)
>10 years	7 (3.2)
No. of Re—Transplantations, *n* (%)	6 (2.7)

Results are given in counts *n* and percentages (%) unless indicated otherwise. AAT, alpha-1 antitrypsin deficiency; LAM, Lymphangioleiomatosis; BCT, Bronchiectasis.

**Table 2 medicina-59-01473-t002:** Number of samples for period 1 vs. period 2 (only USZ + HUG, *n* = 150).

	Time Period 1	Time Period 2
Nasopharyngeal-/pharyngeal-/nasal-swabs (Including COVID-only swabs)	803 (0)	779 (925)
BAL	50	18
COVID-19 swabs	0	146

Results are given in numbers *n*. Time Period 1 (1 March 2019–29 February 2020); Time Period 2 (1 March 2020–28 February 2021). USZ, University Hospital of Zurich; HUG, University Hospitals Geneva; BAL, Bronchoalveolar Lavage.

**Table 3 medicina-59-01473-t003:** Total number of infections, type of virus, and number of hospitalizations for each period.

	Time Period 1	Time Period 2
Total no. of infections for all three centers, *n*	157	53
Co-infections	9	0
incl. SARS-CoV-2	157	71
Type of infections, *n* (%)		
Adenovirus	3 (1.8)	0 (0)
hBocavirus	1 (0.6)	0 (0)
Coronavirus-(229E, HKU1, NL63, OC43)	18 (10.9)	7 (9.9)
hMetapneumovirus	10 (6.0)	0 (0)
Parainfluenza (1–4)	21 (12.7)	0 (0)
hRhino-/Enterovirus	88 (53.0)	43 (60.6)
RSV A/B	8 (4.8)	1 (1.4)
Influenza A	15 (9.0)	2 (2.8)
Influenza B	2 (1.2)	0 (0)
SARS-CoV-2	0 (0)	18 (25.3)
Total no. of hospitalization, *n*	7	0
Influenza A	2	0
hMetapneumovirus	1	0
Parainfluenza	1	0
Parainfluenza/RSV	1	0
Parainfluenza/Rhinovirus	1	0
Rhinovirus	1	0

Results are given in counts *n* and percentages (%); co-infections are counted as 1 infection. Time Period 1 (1 March 2019–29 February 2020); Time Period 2 (1 March 2020–28 February 2021). RSV, Respiratory Syncytial Virus.

**Table 4 medicina-59-01473-t004:** Influenza cases and influenza vaccination rates.

	Time Period 1	Time Period 2
Vaccinated	158	144
Not vaccinated	22	31
Unknown	41	46
Vaccination rate (%)	71.5	65.2
Total Influenza A + B	17	2
Vaccinated	8	1
Not vaccinated	6	1
unknown	3	0

Time Period 1 (1 March 2019–29 February 2020); Time Period 2 (1 March 2020–28 February 2021).

**Table 5 medicina-59-01473-t005:** Number of patients that gave consent but were excluded due to exclusion criteria.

	Time Period 1	Time Period 2
Total deaths	21	14
No. of transplantations	33	34

Time Period 1 (1 March 2019–29 February 2020); Time Period 2 (1 March 2020–28 February 2021).

## Data Availability

Data are available upon reasonable request.

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
