# Peer review of "Impact of SARS-CoV-2-Related Hygiene Measures on Community-Acquired Respiratory Virus Infections in Lung Transplant Recipients in Switzerland"

_medicina, 2023, doi:10.3390/medicina59081473_

Round 1
Reviewer 1 Report
In this very well-written and straightforward study, authors have retrospectively studied the impact of government-imposed hygiene measures for the general population measures on the transmission of community-acquired respiratory virus (CARV) in lung transplant recipients in Switzerland.
Authors report a significant decline in the number of CARV infections in the Swiss lung transplant cohort during the first year of the COVID-19 pandemic. Finding are important and add to the literature in sense that hygiene measures implemented in the population due to the COVID-19 pandemic had a positive impact on the incidence of CARV infections. As told, paper is very well written and I found no major methodologic issue.
Reviewer 2 Report
Major revision is required

Major revision is required
Round 2
Reviewer 2 Report
corrections are completed and It can be accepted now
